# Incorporating Syntactic Knowledge into Pre-trained Language Model using Optimization for Overcoming Catastrophic Forgetting

**Ran Iwamoto**[1,2] **Issei Yoshida**[1] **Hiroshi Kanayama**[1] **Takuya Ohko**[1] **Masayasu Muraoka**[1]

[1]IBM Research - Tokyo, [2]Keio University

ran.iwamoto1@ibm.com,{issei,hkana,ohkot,mmuraoka}@jp.ibm.com

## Abstract

Syntactic knowledge is invaluable information for many tasks which handle complex or long sentences, but typical pre-trained language models do not contain sufficient syntactic knowledge. Thus it results in failures in downstream tasks that require syntactic knowledge. In this paper, we explore additional training to incorporate syntactic knowledge to a language model. We designed four pre-training tasks that learn different syntactic perspectives. For adding new syntactic knowledge and keeping a good balance between the original and additional knowledge, we addressed the problem of catastrophic forgetting that prevents the model from keeping semantic information when the model learns additional syntactic knowledge. We demonstrated that additional syntactic training produced consistent performance gains while clearly avoiding catastrophic forgetting.

## 1 Introduction

Pre-trained language models are commonly used and improve the performance of a variety of application tasks. It has been shown that those models roughly capture syntactic knowledge (Chi et al., 2020), such as dependency labels, but they lack some syntactic knowledge such as dependency distance and head token required for application tasks (Xu et al., 2021).

Recent studies have shown that incorporating syntactic knowledge into the models further improves the performance of language understanding (Zhang et al., 2020) and translation tasks (Bugliarello and Okazaki, 2020) by adding other modules to the core model in the training and application phases.

In contrast, Tian et al. (2022) and Yang and Wan (2022) take approaches to embed syntactic knowledge into a model with keeping model structures. In this paper we follow this approach (henseforth we call it *additional syntactic training*), because

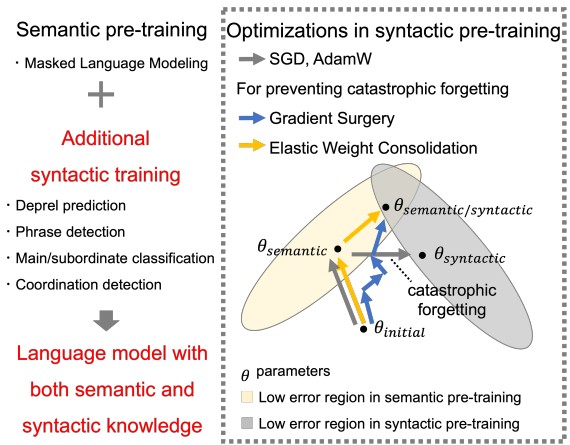

Figure 1: An overview of additional syntactic training. Syntactic knowledge is added to the language model while preserving original semantic knowledge in the model using proper optimization functions.

the models empowered with syntax knowledge and retaining semantic knowledge can be easily applied to downstream tasks and, the methodology is useful for injecting other knowledge into the language model to create knowledge-enhanced models such as domain-specific ones.

Figure 1 depicts our approach and techniques to obtain desirable language models. We start with a language model such as BERT (Devlin et al., 2019) which was pre-trained with a large amount of unlabeled text data and self-supervised tasks such as Masked Language Modeling. Such a model contains semantic knowledge but does not have explicit syntactic information, thus we enhance the model with labeled data with syntactic information.

One major issue of additional syntactic training is that we need to find a good balance between the original and additional knowledge for improving performance of downstream tasks. Most of the previous works have selected dependency structure predictions as the task to learn syntactic knowledge. However, there are many other specific syntactic notions, such as phrase structures and coordina-

tions from which the additional training may benefit, and it is non-trivial to identify the best ones.

Another big issue in balancing the model and additional knowledge is *catastrophic forgetting* (French, 1999; Kemker et al., 2018), where additional training causes a significant loss of important knowledge embedded in the original model. To tackle this problem, we exploit multiple optimizers and quantitatively observe the balance of capabilities to predict semantic and syntactic features. To the best of our knowledge, our study is the first to analyze how both the types of additional syntactic training and its optimizations affect downstream tasks.

Our contributions in this paper are as follows:

- Designed additional syntactic training with four syntactic tasks that can help a language model solve downstream tasks from different syntactic perspectives.

- Exploited optimization functions that prevent catastrophic forgetting during the additional syntactic training, and discovered the trade-off between retention of original knowledge and enhancement with additional knowledge.

- Demonstrated that the syntactically enhanced models improved performances on CoLA, RTE, MRPC and key phrase extraction tasks, with qualitative and quantitative discussions to find the syntactic perspectives that contribute to each task.

## 2 Additional Syntactic Training

In additional syntactic training, models mainly learn syntactic relationships between two tokens. There have been proposed tasks such as predicting dependencies (Wang et al., 2021a), predicting parent-child, sibling, or cousin relationships (Tian et al., 2022), and dependency distance. The dependency structure encompasses a variety of word relationships, a number of which are important in application tasks.

For example, in noun phrase extraction (Gu et al., 2021) and sentiment analysis (Kanayama and Iwamoto, 2020), the relationship between main and subordinate clauses and parallel (coordinate) structures are not handled well. To reflect the syntactic knowledge missing and required for downstream tasks in pre-trained models, we need to perform more diverse additional syntactic training tasks.

| | Text | | Syntax | | Syntax Label | | |
|---|---|---|---|---|---|---|---|
| ID | form | PoS | Head | deprel | phrase | main/sub | coord |
| 1 | We | PRON | 3 | nsubj | nsubj | main | other |
| 2 | still | ADV | 3 | advmod | advmod | main | other |
| 3 | have | VERB | 0 | root | root | main | other |
| 4 | the | DET | 5 | det | obj | main | child |
| 5 | traders | NOUN | 3 | obj | obj | main | conj |
| 6 | and | CCONJ | 7 | cc | obj | main | cc |
| 7 | books | NOUN | 5 | obj | obj | main | conj |
| 8 | that | PRON | 10 | obj | obj | sub | other |
| 9 | you | PRON | 10 | nsubj | nsubj | sub | other |
| 10 | provided | VERB | 5 | acl | acl | sub | other |
| 11 | last | ADJ | 12 | amod | obl | sub | other |
| 12 | week | NOUN | 10 | obl | obl | sub | other |
| 13 | . | PUNCT | 3 | punct | punct | main | other |

Table 1: An example of additional syntactic training tasks. With Tian et al.'s (2022) system, the model learns Syntax Head and Label simultaneously. Gray areas denote phrases, main clauses, and coordination structures.

We therefore develop four pre-training tasks to predict various syntactic items. Each task predicts specific syntactic information: 1) deprel prediction, 2) phrase detection, 3) main/subordinate classification, and 4) coordination detection. We investigate which task would be effective on applications.

### 2.1 Additional Syntactic Training System

Here, we describe our method for additional syntactic training. We customize Tian et al.'s (2022) system to learn syntactic information.

The system is trained on two tasks in parallel: dependency masking (DM), which predicts whether there is a dependency between any pair of words in a sentence, and masked dependency prediction (MDP), which predicts the type of dependency relationships. We replace the input for MDP with one of four pre-training tasks.

### 2.2 Syntactic Tasks

Table 1 illustrates the four variations of our pre-training tasks. Syntax Head conveys the form of dependency tree, which is used for DM in the system described in Section 2.1 and fixed in this paper. As MDP, we vary the viewpoint of syntax classification with the following four labeling tasks.

**Deprel prediction (deprel)** The model predicts dependency labels in the same way as UD (Nivre et al., 2020) parsing in the task. For additional syntactic training tasks, we position this task as a baseline.

**Phrase detection (phrase)** In this task, a model predicts the relationship between phrases (the deprel label of the head token in the phrase). It is inspired by Basirat and Nivre's (2021) work, and we

follow their definition of phrase-like units. They defined a phrase as the block connected by seven UD functional relations, such as a determiner and case marker. Their model achieved high performance in dependency parsing by phrase-level pre-training.

**Main/subordinate classification (main/sub)** The task predicts two labels to classify clauses into main and sub. We refer to Nikolaev and Pado's (2022) method to investigate whether a pre-trained model can detect subordinate clauses.

**Coordination detection (coord)** It aims to predict parallel structures like "A and B" where A and B can be tokens, phrases, or sentences. Specifically, it predicts the labels of A and B's head tokens (conj), their child tokens (child), parallel conjunctions such as "and" and "but" (cc), and other tokens (other).

# 3 Optimization Functions to Prevent Catastrophic Forgetting

This section describes two optimization functions that prevent catastrophic forgetting for keeping original semantic knowledge during additional syntactic training. When the model has already been trained for a specific task and is then trained for another task, the performance of the previous task significantly decreases. This is called catastrophic forgetting (French, 1999; Kemker et al., 2018).

Although large-scale pre-trained models are known to be relatively resilient to catastrophic forgetting (Ramasesh et al., 2022), the problem remains when using small models such as BERT (Devlin et al., 2019). Since BERT is still often used in the real world task due to computational resources, a variety of methods are developed to prevent catastrophic forgetting even in pre-trained models such as decreasing the learning rate (Kar et al., 2022).

Among these methods, we used Gradient Surgery (Yu et al., 2020), an optimization function known in the domain of multi-task learning, and Elastic Weight Consolidation (Kirkpatrick et al., 2017), an optimization function commonly used in continual learning.

## 3.1 Gradient Surgery (GS)

Gradient Surgery is an optimization function for solving gradient conflicts in multi-task learning. It first computes the gradients for each task involved in the multi-task learning. We then discard adversarial elements of the gradients that conflict with

each other. After that, we sum up the resultant gradients to obtain a single gradient vector. For example, given two tasks A and B, if their gradient vectors $\boldsymbol{g}_A$ and $\boldsymbol{g}_B$ are in opposite directions (i.e., if the cosine similarity of $\boldsymbol{g}_A$ and $\boldsymbol{g}_B$ is negative), one gradient $\boldsymbol{g}_A$ is projected onto the orthogonal plane of the other gradient $\boldsymbol{g}_B$ as follows:

$$\boldsymbol{g}_A = \boldsymbol{g}_A - \frac{\boldsymbol{g}_A \cdot \boldsymbol{g}_B}{\|\boldsymbol{g}_B\|^2}\boldsymbol{g}_B,$$

where $\|\boldsymbol{x}\|$ denotes L2-norm of $\boldsymbol{x}$.

## 3.2 Elastic Weight Consolidation (EWC)

Elastic Weight Consolidation is a continuous learning optimization function in which tasks are learned in sequence. Given two tasks, A and B, the model first searches for the optimal solution for task A, and then for the parameters that perform well in both tasks A and B. In EWC, when the model is fine-tuned for B after A, the important parameters of the task A are updated as little as possible, while the less important parameters of task A are updated with larger weights. We use the Fisher information matrix $F$ (Pascanu and Bengio, 2013), which is a diagonal matrix, to reduce the computational cost and to utilize characteristics of $F$ (Pascanu and Bengio, 2013), though it gives an approximation of the optimal parameters $\mathcal{L}(\theta)$ for task A. The loss function of EWC $\mathcal{L}(\theta)$ is as follows:

$$\mathcal{L}(\theta) = \mathcal{L}_B(\theta) + \sum_i \frac{\lambda}{2}F_i\big(\theta_i - \theta_{A,i}^*\big)^2,$$

where $\mathcal{L}_B(\theta)$ is the task B's loss, $\lambda$ sets how important the task A is compared with the task B, and $i$ labels each parameter.

In the rest of this paper, we show our experimental results of three different tasks; additional syntactic training (Section 4), the GLUE benchmark (Section 5), and key phrase extraction (Section 6) and explore how syntactic information and different optimization functions affect the pre-trained model itself and downstream tasks.

# 4 Preliminary Experiment: Additional Syntactic Training

Here, we first show how well pre-trained models with semantic information can learn specific syntactic information by additional syntactic training tasks (See Section 2.2 for detail) using various optimization functions while preserving semantic information.

| | train | dev | test |
|---|---|---|---|
| original | 12543 | 2001 | 2077 |
| after filtering | 10271 | 1471 | 1485 |

Table 2: Number of sentences in the UD-EWT corpus and after filtering for additional syntactic training.

## 4.1 Settings

**Data**  The data for syntactic training was generated from UD-EWT Ver 2.10 [1]. The sentences satisfying any of the following three conditions were removed as noise: 1) sentences with less than five tokens, 2) sentences containing foreign languages tagged as X, and 3) sentences containing undefined dependency relations labeled as dep. The number of sentences used in the experiment was the same across all tasks since these treatments were applied to all pre-training tasks. Table 2 shows the statistics.

**System**  We used the syntactic pre-training system described in Section 2. We adopted the pre-trained `bert-base-cased` model downloaded from the Hugging Face transformers library (Wolf et al., 2020) in the following experiments. The models trained by each of the four additional syntactic training methods described in Section 2.2 were compared with the original `bert-base-cased` model. Multiple learning rates {1e-4,1e-5,1e-6} and multiple numbers of max epochs {50,70,100} were tested.

We experimented with four optimization functions: GS, EWC, AdamW, and SGD. GS and EWC prevent catastrophic forgetting as described in Section 3. We used GS implemented by its authors (Tseng, 2020) and EWC by ours, respectively. For comparison, AdamW (Loshchilov and Hutter, 2019) and SGD (Robbins and Monro, 1951) from the official PyTorch library (Paszke et al., 2019) were used.

Models using GS are trained with gradients computed from a small amount of text data used for Masked Language Modeling (we call them MLM gradients afterward), while models using EWC only use MLM gradients for determining the importance of parameters, not for parameter updating. AdamW and SGD do not consider MLM gradient information. For training with GS and EWC, 100 sentences were randomly selected from Wiki-

[1] https://github.com/UniversalDependencies/UD_English-EWT

| syntactic training | optimization function | F1 | PPL ($\downarrow$) |
|---|---|---|---|
| none | - | - | 29.10 |
| deprel | AdamW | 96.22 | 702.87 |
| | SGD | 96.13 | 32392.10 |
| | GS | 95.43 | **10.44** |
| | EWC | 95.13 | 25.63 |
| phrase | AdamW | 97.25 | 1360.61 |
| | SGD | 95.62 | 2587.11 |
| | GS | 95.78 | **9.68** |
| | EWC | 95.58 | 24.17 |
| main/sub | AdamW | 98.35 | 720.49 |
| | SGD | 97.74 | 1471.63 |
| | GS | 94.70 | **11.16** |
| | EWC | 95.29 | 23.59 |
| coord | AdamW | 94.68 | 2971.79 |
| | SGD | 89.55 | 1412.65 |
| | GS | 91.33 | **9.56** |
| | EWC | 91.00 | 23.78 |

Table 3: Comparison with different syntactic tasks and optimization functions in additional syntactic training. F1 score is the average between syntax head F1 and syntax label F1. The row "none" means the baseline model without additional syntactic training.

Text2 (Merity et al., 2017) for each syntactic training step, in which MLM gradients were computed.

It may be worthwhile to include MLM pre-training using AdamW/SGD on the corpus used for additional syntactic training as comparisons. Although it is already known that AdamW (or Adam) and SGD can cause catastrophic forgetting in both continual learning and multi-task learning. Also in multi-task optimization, existing studies such as GS, Gradient Vaccine (Wang et al., 2021b), and GradNorm (Chen et al., 2018) showed that models using Adam or SGD do not perform well.

**Evaluation**  We measured precision, recall, and F1 score in the additional syntactic training tasks. To evaluate whether the model retains the original semantic knowledge, we used the perplexity (PPL) (Jelinek et al., 1977) score of the MLM.

## 4.2 Results

Table 3 reports the highest average F1 scores of Syntax Head and Syntax Label predictions. There was a significant difference between different optimization functions in the PPL measure of MLM. AdamW and SGD focus only on improving the performance of the target task (in this case, syntactic

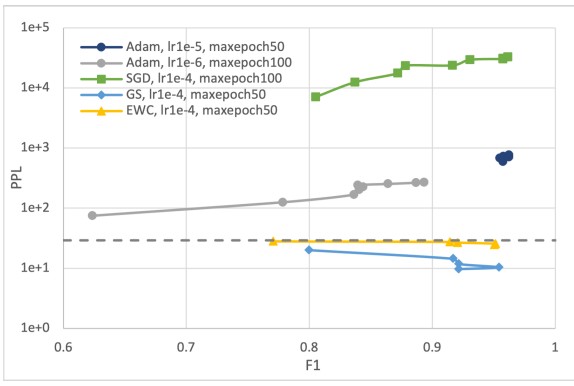

Figure 2: Relationships between syntactic pre-training F1 score and PPL. The checkpoints at every 10 epochs are plotted in syntactic pre-training phase. The dotted line represents PPL of bert-base without additional syntactic training.

training), thus they achieve high F1 scores in the target task with significantly degrading the previous task, which was indicated by high PPL scores (catastrophic forgetting). Much lower PPL scores in GS and EWC indicate that they retain semantic knowledge gained before additional syntactic training.

All syntactic pre-trained models achieved over 91% average F1 scores except for the model trained with SGD on the coord task, indicating that most models have sufficiently learned syntactic information. Especially, there is a small difference in F1 scores between models trained with AdamW and other optimization functions in the deprel task.

We explain why we use GS and EWC in additional training and downstream tasks. The relationships between syntactic pre-training F1 score and PPL are shown in Figure 2. The figure displays the models that achieved the highest F1 score for each optimization function, as well as the model with the smallest learning rate using Adam for reference. We plot checkpoints at every 10 epochs in syntactic additional training phase. GS and EWC can keep their PPLs below the bert-base PPL throughout the learning process, whereas Adam and SGD have very high PPL during learning (catastrophic forgetting). From this fact, we conclude GS and EWC are suitable for retaining both semantic and syntactic knowledge.

## 5   Experiment 1: GLUE

We investigate the effects of syntactic pre-trained models to multiple downstream tasks using GLUE benchmark (Wang et al., 2018). GLUE has three types of tasks: single sentence tasks (CoLA, SST-2), similarity and paraphrase tasks (MRPC, QQP and STS-B) and natural language inference tasks (MNLI, QNLI, RTE). We conducted detailed experiments on three tasks for which syntactic information seems to be important in solving the tasks: CoLA (Warstadt et al., 2019), RTE (Dagan et al., 2006), and MRPC (Dolan and Brockett, 2005). Experiments on other tasks that did not required significant syntactic knowledge are shown in Appendix. From experiments of the GLUE subset, it can be concluded that additional syntactic training is beneficial for tasks involving syntactic knowledge, as opposed to the tasks mentioned in the Appendix A, which do not appear to require such knowledge.

### 5.1   Settings

We evaluated on the official data splits provided by GLUE. The details and evaluation metrics of each task are described in Appendix A. As the overall score for our methods, we took a macro-average of scores for the tasks. The number of epoch was set to 5, and other hyperparameters were set to the default values specified in the code provided by Hugging Face[2].

We trained models for each setting (of four optimizers and four syntactic tasks). The training is conducted over three learning rates and three of max epochs described in Section 4.1, saving the checkpoints at every 10 epochs in syntactic pre-training phase. We then trained all of these models for the application tasks. Therefore, we trained approximately 1000 syntactic pre-trained models: 4 optimizers × 4 syntactic tasks × 3 learning rates × 22 checkpoints[3].

For each pair of pre-training task and optimizer, we selected the best model in terms of the average score of each GLUE task. To minimize the computational cost, we adopted the optimal hyperparameters used in a previous study (Wang et al., 2018). The reason we chose this approach was to determine how much knowledge from MLM and syntactic structures is needed to achieve good performance in application tasks.

Regarding evaluation, we considered that the evaluation on dev data is appropriate for this purpose to keep the test data unbiased to our tuning, following the policy by Suzuki et al. (2023).

---

[2]https://github.com/huggingface/transformers/blob/main/examples/pytorch/text-classification/run_glue.py

[3]Every 10 epochs for max epochs of 50, 70, 100.

| Syntactic Training | | GLUE tasks | | | | Pre-train info | |
|---|---|---|---|---|---|---|---|
| | | CoLA | RTE | MRPC | Avg. | lr | epoch |
| Task | Optimizer | (mc) | (acc) | (acc / F1) | | | |
| none | - | 60.10 | 64.62 | 84.56 / 89.23 | 70.54 | - | - |
| deprel | AdamW | 57.60 | 66.43 | 78.19 / 85.85 | 68.68 | 1e-5 | 10/70 |
| | SGD | 57.80 | 67.15 | 85.05 / 89.78 | 70.79 | 1e-5 | 10/50 |
| | GS | 59.07 | 67.51 | **86.02** / 89.98 | 71.58 | 1e-4 | 20/100 |
| | EWC | 60.57 | 66.07 | 85.54 / 89.92 | 71.45 | 1e-4 | 30/70 |
| phrase | AdamW | 55.98 | 67.15 | 78.68 / 86.21 | 68.53 | 1e-5 | 20/70 |
| | SGD | 58.30 | 67.87 | 84.31 / 89.33 | 71.00 | 1e-5 | 10/50 |
| | GS | 60.86 | 64.98 | 84.56 / 89.23 | 70.91 | 1e-5 | 20/70 |
| | EWC | 60.32 | 67.15 | 86.28 / **90.54** | 71.96 | 1e-5 | 20/50 |
| main/sub | AdamW | 55.73 | 66.07 | 77.94 / 85.44 | 67.83 | 1e-5 | 40/70 |
| | SGD | 57.31 | 67.51 | 85.29 / 89.90 | 70.81 | 1e-5 | 10/50 |
| | GS | 59.10 | 67.51 | 85.29 / 89.83 | 71.39 | 1e-5 | 20/70 |
| | EWC | 60.09 | 68.23 | 84.80 / 89.49 | 71.82 | 1e-5 | 20/70 |
| coord | AdamW | 58.29 | 63.90 | 77.45 / 85.16 | 67.83 | 1e-5 | 60/70 |
| | SGD | 59.31 | 66.07 | 83.82 / 88.81 | 70.56 | 1e-5 | 20/70 |
| | GS | 59.56 | **68.59** | 85.54 / 90.02 | 71.98 | 1e-5 | 60/70 |
| | EWC | **61.10** | 68.23 | 85.05 / 89.61 | **72.22** | 1e-5 | 30/50 |

Table 4: Performance comparison of three GLUE tasks based on additional syntactic training and optimizers. The highest scores for each additional syntactic training are underlined, and those for each GLUE task are shown in bold. The row "none" means the baseline model without additional syntactic training.

| Optimizer | Average | | |
|---|---|---|---|
| | CoLA | RTE | MRPC |
| AdamW | 56.90 | 65.88 | 78.06 / 85.67 |
| SGD | 58.18 | 67.15 | 84.62 / 89.46 |
| GS | 59.65 | 66.79 | **85.48 / 89.90** |
| EWC | **60.52** | **67.42** | 85.42 / 89.89 |

Table 5: Comparison of model average for each optimizer in three GLUE tasks.

| Syntactic Task | Average | | |
|---|---|---|---|
| | CoLA | RTE | MRPC |
| deprel | 58.76 | 66.43 | **83.82 / 89.02** |
| phrase | 58.87 | 66.79 | 83.46 / 88.83 |
| main/sub | 58.06 | **67.33** | 83.33 / 88.66 |
| coord | **59.56** | 66.70 | 82.97 / 88.40 |

Table 6: Comparison of model average for each additional syntactic training in three GLUE tasks.

## 5.2 Results

The results are shown in Table 4. The right block is the performance in the pre-training with lr and epoch on syntactic tasks and PPL on MLM task.

The models using GS and EWC, which were introduced to prevent catastrophic forgetting, achieved higher scores, as expected, than those using AdamW and SGD in CoLA and MRPC tasks. Among the 16 combinations of syntactic tasks and optimizers, coord with EWC achieved the highest average score (72.22), and most of other cases showed scores above the baseline (70.54) except for those optimized with AdamW. In the following, we compare the performance of each optimizer and each pre-training task by taking the average score on each feature.

**Comparison of optimizers** Table 5 shows the scores of each optimizer averaged over four syntactic training tasks in Table 4. The model pre-trained with EWC achieves higher scores on two tasks, and the model pre-trained with GS has similar scores. That is, regardless of the type of syntactic training task, learning syntax using GS and EWC is effective to enhance the model performance.

As shown in the right block of Table 3 GS and EWC preserve semantic information with small sacrifice of additional syntactic training scores. In the trade-off relationship between these two, Table 5 shows the strength of GS and EWC in downstream tasks, models with a good balance between semantic information and syntactic information achieved high performance.

| Syntactic Training | | Key Phrase Extraction | | | Pre-train Info | |
|---|---|---|---|---|---|---|
| Task | Optimizer | precision | recall | F1 | lr | epoch |
| none | - | 59.71 | 74.11 | 66.14 | - | - |
| deprel | AdamW | 62.41 | 75.22 | 68.22 | 1e-5 | 10/50 |
| | SGD | 62.50 | 74.78 | 68.09 | 1e-5 | 60/100 |
| | GS | 62.24 | 75.30 | 68.15 | 1e-4 | 60/70 |
| | EWC | 61.43 | 75.97 | 67.93 | 1e-5 | 30/60 |
| phrase | AdamW | 62.52 | 75.07 | 68.22 | 1e-5 | 60/100 |
| | SGD | 62.09 | 76.64 | 68.60 | 1e-4 | 10/70 |
| | GS | 62.52 | 76.34 | **68.74** | 1e-6 | 60/100 |
| | EWC | 61.77 | 75.97 | 68.13 | 1e-5 | 40/100 |
| main/sub | AdamW | 62.40 | 75.07 | 68.15 | 1e-6 | 30/100 |
| | SGD | 61.58 | 75.60 | 67.8 | 1e-4 | 60/70 |
| | GS | 62.04 | 75.89 | 68.27 | 1e-4 | 10/50 |
| | EWC | 61.12 | 75.45 | 67.53 | 1e-5 | 20/70 |
| coord | AdamW | 61.68 | 76.41 | 68.26 | 1e-5 | 10/70 |
| | SGD | 62.89 | 75.52 | 68.63 | 1e-4 | 10/70 |
| | GS | 61.42 | 75.82 | 67.87 | 1e-6 | 60/100 |
| | EWC | 62.03 | 75.60 | 68.14 | 1e-5 | 40/100 |

Table 7: Performance comparison of key phrase extraction task.

**Comparison of additional syntactic training**
Table 6 shows the scores of each pre-training tasks averaged over the four optimizers in Table 4. A variety of syntactic information contributed to the performance improvement. In the CoLA task, coord achieved the highest average score. We explain this using the data grammatical features. Warstadt and Bowman (2019) compared CoLA performance for sentences containing major features using dev data. They defined 13 grammatical major features and further divided into 59 minor features, and coordination is the most frequent minor feature in a major feature. They found that in the BERT model, Matthew's correlation for sentences containing the major feature was the second lowest among all major features. This indicates that the CoLA task performance was improved because models captured the coordination structures through additional additional syntactic training.

The RTE task contains many long sentences, and we consider that the main/sub pre-trained model handled long sentences better than the other models since the model can capture subordinate structures.

For the MRPC task, deprel achieved the highest average score. Perez et al. (2021) mentioned that MRPC requires information with various PoS tags via experiments in which models are trained by masking words of a specific PoS tag in each GLUE tasks. As a counter example, for SST-2, sentiment analysis task, adjectives are much more effective than other PoS. The deprel task helped the model learn more comprehensive syntactic information than the other tasks, which explains our result.

# 6 Experiment 2: Key Phrase Extraction

In addtion to binary classification tasks in GLUE, we conducted another experiment on the *key phrase extraction task* (Gu et al., 2021). It is a task to detect domain-specific phrases in sentences, and is considered a good workbench to test the ability to extract syntactically correct sequences as an application close to real-world use cases.

## 6.1 Settings

We used Gu et al.'s (2021) method in our experiments. They trained a phrase tagger using language models by generating silver labels from unlabeled text and using neither external knowledge base nor dictionaries. We modified it to work with BERT since their code worked only with models using byte pair encoding tokenizers. The number of epochs was set to 100. Other hyperparameters followed the default settings in the code.

In experiments by Gu et al. (2021), datasets KP20k (Meng et al., 2017) and KPTimes (Gallina et al., 2019) are used. Since we evaluated many models, we used a small version of the KP20k provided in their code to reduce computational time.

| Statistics | train | test |
|---|---|---|
| # document | 10000 | 1000 |
| # sentence | 81340 | 8144 |
| # words per sentence | 21.50 | 21.48 |

Table 8: Statistics in KP20k small datasets.

| Syntactic | Average | | |
|---|---|---|---|
| Task | prec | rec | F1 |
| deprel | 62.15 | 75.32 | 68.10 |
| phrase | **62.23** | **76.01** | **68.42** |
| main/sub | 61.79 | 75.50 | 67.96 |
| coord | 62.01 | 75.84 | 68.22 |

Table 9: Comparison of average scores of syntactic pre-trained models in the key phrase extraction.

KP20k is a dataset created from titles, abstracts, and key phrases of scientific articles in computer science. The data statistics are shown in Table 8. We selected the models in the same manner as we did in Section 5.1 for the three GLUE tasks.

## 6.2 Results

The results are shown in Table 7. As well as in the experiments in Section 5, we fine-tuned the syntactic pre-treined models for this task. All the syntactic models achieved higher F1 scores than the baseline by more than 1.4 points, regardless of the optimizers. This indicates that a variety of syntactic information contributes to improving key phrase extraction, and the effect is bigger than in the binary classification tasks in GLUE.

Table 9 shows the average scores over for optimizers. The results supports that phrase was the best in all four metrics. As shown in Table 1, phrase has an intermediate granuality between deprel and main/sub, and it is intuitive that this matches well the syntactic knowledge required in the key phrase extraction task.

## 6.3 Case Study

We present qualitative analysis for showing how added syntactic knowledge is reflected in the model extractions. We observe the difference between the baseline model without additional syntactic training and pre-trained model with the phrase task. The phrase pre-trained model using GS achieved the best F1 score as observed in Section 6.2.

In Figure 3, we can find that the baseline model incorrectly detected "reduce business costs" as a key phrase, because it failed to recognize "reduce"

**[Gold Answer]**
In addition to recycling , use of a RFID system can also reduce **business costs** , by identifying the position of goods and picking carts .

**[Baseline (any syntactic pretraining)]**
In addition to recycling , use of a RFID system can also **reduce business costs** , by identifying the position of goods and picking carts .

**[Syntactic pretraining (phrase task, using GS)]**
In addition to recycling , use of a RFID system can also reduce **business costs** , by identifying the position of goods and picking carts .

Figure 3: Phrase extraction 1. The same phrase extractions among gold answer and two methods are blue-underlined, different phrase extractions are written in bold yellow. Wrong output: "reduce business costs."

**[Gold Answer]**
A geometric-based method for recognizing overlapping polygonal-shaped and semi-transparent particles in gray tone images .

**[Baseline (any syntactic pretraining)]**
A geometric-based method for **recognizing overlapping** polygonal-shaped and semi-transparent particles in gray tone images .

**[Syntactic pretraining (phrase task, using GS)]**
A geometric-based method for recognizing overlapping polygonal-shaped and semi-transparent particles in gray tone images .

Figure 4: Phrase extraction 2. Wrong output: "recognizing overlapping."

as a verb in this sentence. On the other hand, phrase pre-trained model successfully detected "business costs". In Figure 4, the baseline model incorrectly detected "recognizing overlapping" as a phrase. The phrase pre-trained model successfully avoided a phrase extraction error caused by a word whose grammatical behavior may be confused, in this case, between gerund and present progressive.

## 7 Conclusion

This paper investigated the incorporation of syntactic knowledge into a pre-trained BERT model by additional training. We designed four additional syntactic training tasks: deprel prediction, phrase detection, main/subordinate classification, and coordination detection. We utilized GS and EWC optimizations for preserving semantic knowledge.

The experimental results showed improvements of the models empowered with syntactic knowledge in the downstream tasks, and supported our intuition that each task requires different syntactic perspectives for more accurate predictions.

## Limitations

Our main limitation is that our method is evaluated on English-only datasets including GLUE benchmarks and key phrase extraction datasets. The applicability of our method to other languages can be investigated by evaluating, for example, a multilingual version of the GLUE task (Liang et al., 2020; Kurihara et al., 2022).

An additional limitation comes from the pre-trained model. We used a pre-trained BERT model with 110M parameters generated for general purpose tasks. The investigation methods in this paper can be applied to other language models such as RoBERTa (Liu et al., 2019), though they may show different trends through further experiments.

## Ethics Statement

Our method belongs to additional language model pre-training. We used publicly available codes/datasets such as licensed under CC BY-SA 4.0 and MIT to ensure reproducibility of our experimental results. Using an efficient optimization function rather than many parameter tuning implementations to prevent catastrophic forgetting can help reduce the carbon footprint of model training.

Our method uses pre-trained language models that may contain gender or abusive data bias. We need to be careful when using the model to prevent potential biases in application task outcomes.

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

# A  GLUE task explanation

We describe the details of dataset in our experiments GLUE (Wang et al., 2018), the General Language Understanding Evaluation benchmark. GLUE is the common benchmark for language comprehension. The licence is under CC-BY-4.0.

## A.1  Syntactic-related Tasks

Here we explain three GLUE tasks in which additional syntactic training would be highly useful.

**CoLA**  The Corpus of Linguistic Acceptability (CoLA) (Warstadt et al., 2019) is composed of English sentences extracted from books and journal articles focused on linguistic theory. Each sentence is annotated to indicate whether it is grammatically acceptable in English. To evaluate performance, we utilize the Matthews correlation (mc), a metric for evaluating binary classification performance, which accounts for class imbalance and ranges from -1 to 1, with 0 representing random guessing. For evaluation purposes, we utilize the standard test set and obtained private labels from the original authors same as official GLUE evaluation.

**RTE**  The Recognizing Textual Entailment (RTE) (Dagan et al., 2006) datasets come from a series of annual textual entailment challenges. The examples in the datasets are created using news articles and text from Wikipedia. To ensure uniformity, all datasets are splitted into a two-class. In three-class datasets, we combine the "neutral" and "contradiction" classes into the category of "not entailment."

**MRPC**  The Microsoft Research Paraphrase Corpus (MRPC) (Dolan and Brockett, 2005) consists of pairs of sentences extracted from online news sources. Each pair has been annotated by humans to indicate whether the sentences are semantically equivalent. Due to the imbalance in class distribution (68% positive), we report both accuracy and F1 score (standard practice).

## A.2  Syntactic-unrelated Tasks

The results of those tasks shown in appendix.

**SST-2**  The Stanford Sentiment Treebank-2 (SST-2) (Socher et al., 2013), developed by Socher et al. in 2013, comprises sentences extracted from movie reviews along with human annotations indicating their sentiment. The objective is to determine the sentiment of a given sentence. We employ a binary classification scheme, distinguishing between positive and negative sentiments, and utilize only the labels assigned at the sentence level.

**QQP**  The Quora Question Pairs2 dataset (QQP) [4] consists of a set of question pairs gathered from the Quora website, where users ask and answer questions. The objective is to determine whether a given pair of questions are semantically equivalent. Similar to MRPC, the distribution of classes in the QQP dataset is imbalanced, with a majority of 63% negative instances. Hence, we report both accuracy and F1 score for evaluation.

**STS-B**  The Semantic Textual Similarity Benchmark (Cer et al., 2017) consists of various sentence pairs obtained from news headlines, video and image captions, and natural language inference data. Each pair has been assessed by humans and assigned a similarity score ranging from 1 to 5. The objective of the task is to predict these scores. In our analysis, we evaluate the performance using Pearson correlation coefficients (pc).

**MNLI**  The Multi-Genre Natural Language Inference dataset (MNLI) (Williams et al., 2018) is a collection of sentence pairs of textual entailment. The task involves predicting whether the premise sentence entails, contradicts, or is neutral to the hypothesis sentence. The sentences are extracted from transcribed speech, fiction, and government reports and so on. Evaluation is conducted on both the matched (in-domain,m) and mismatched (cross-domain,mm) sections.

**QNLI**  The Stanford Question Answering Dataset (QNLI) (Rajpurkar et al., 2016) is a collection of question-paragraph pairs designed for question-answering tasks. The dataset includes questions generated by annotators and paragraphs sourced from Wikipedia, with the answer to each question located within one of the paragraph's sentences. To transform the task into sentence pair classification, the pairs are created between each question and every sentence within the corresponding context. The objective is to determine whether the context sentence contains the answer to the question. We evaluate accuracy on this task.

---

[4]https://www.quora.com/profile/Ricky-Riche-2/First-Quora-Dataset-Release-Question-Pairs

| Syntactic Training | | GLUE Tasks | | | | |
| Task | Optimizer | SST-2 (acc) | QQP (acc / F1) | STS-B (pc) | MNLI m/mm m/mm (acc) | QNLI (acc) |
|---|---|---|---|---|---|---|
| none | - | **92.66** | 90.98 / 87.78 | 89.08 | **83.71** / 83.34 | 90.55 |
| deprel | AdamW | 91.74 | 90.88 / 87.66 | 88.00 | 83.29 / 83.80 | **90.94** |
| | SGD | 92.20 | **91.05** / **87.89** | 89.27 | 83.52 / 84.18 | 90.87 |
| | GS | 92.32 | 90.85 / 87.62 | **89.36** | 83.28 / 83.70 | 90.68 |
| | EWC | 91.51 | 90.88 / 87.65 | 89.21 | 83.12 / 83.69 | 90.55 |
| phrase | AdamW | 91.17 | 90.48 / 87.18 | 85.49 | 82.36 / 83.01 | 90.17 |
| | SGD | 91.86 | 90.72 / 87.44 | 87.17 | 83.11 / 83.45 | 90.81 |
| | GS | 92.43 | 90.82 / 87.55 | 88.56 | 83.53 / 83.86 | 90.26 |
| | EWC | 92.09 | 90.91 / 87.71 | 89.26 | 82.93 / 83.57 | 90.37 |
| main/sub | AdamW | 89.68 | 90.10 / 86.65 | 85.42 | 81.71 / 82.45 | 89.11 |
| | SGD | 91.97 | 90.84 / 87.59 | 87.57 | 83.31 / 83.83 | 90.19 |
| | GS | 92.09 | 90.94 / 87.73 | 89.35 | 83.61 / 83.73 | 90.37 |
| | EWC | 91.97 | 90.90 / 87.68 | 89.15 | 83.47 / 84.06 | 90.65 |
| coord | AdamW | 90.60 | 89.80 / 86.24 | 82.67 | 83.11/ 83.45 | 87.99 |
| | SGD | 92.20 | 90.92 / 87.71 | 88.15 | 83.67 / **84.19** | 90.61 |
| | GS | 91.86 | 90.84 / 87.60 | 89.27 | 83.45 / 83.55 | 90.32 |
| | EWC | 91.86 | 90.98 / 87.79 | 89.13 | 83.43 / 83.66 | 90.26 |

Table 10: Performance comparison of five GLUE tasks based on additional syntactic training and optimizers. The highest scores for each additional syntactic training is underlined, and those for each GLUE task are shown in bold. The row "none" means the baseline model without additional syntactic training.

## B GLUE results

In the downstream tasks listed below, the models without syntactic information performed higher than the models with it. Our assumption was supported by the results in Table 10, where most of additional syntactic training did not contribute to the scores as we have seen for our focused tasks in Table 4.

**SST-2** We found that the model without additional syntactic training showed the highest score. It can be argued from these results that syntactic knowledge is rather a handicap for sentiment analysis of movie reviews. Among the models with additional syntactic training, the model with GS scores higher since it does not lose semantics.

**QQP** QQP (question pair dataset) contains more sentence pairs that paraphrase or have a completely different meaning than sentence pairs that have been syntactically rewritten. Therefore, there is little difference between the models with and without additional syntactic training, and the models with additional syntactic training score slightly lower. This result suggests that syntactic knowledge is not necessary to solve this task.

**STS-B** Models pre-trained on the GS tend to have higher accuracy. To explain this, we show an example of the data (news headline). For example, a pair of similar sentences "A plane is taking off." / "An air plane is taking off." has the similar syntactic structure and difference is only an adjective. This is because we believe that the model retaining semantic knowledge performed better.

**MNLI** For the matched class (two sentences are sampled from the same source), the models without additional syntactic training achieved the highest accuracy, while for the mismatched class (two sentences are from different sources), most models with additional syntactic training achieved better performance than those without additional syntactic training. We could not identify the reason for the difference in accuracy between different classes.

**QNLI** The models with additional syntactic training achieved higher accuracy than those without additional syntactic training, with a trend towards higher accuracy on the deprel task. This indicates that the deprel task is effective at capturing the overall structure of relatively long answer sentences. We could not observe any differences in scores between the different optimizers.