# OpenReview forum: "Incorporating Syntactic Knowledge into Pre-trained Language Model using Optimization for Overcoming Catastrophic Forgetting"
_EMNLP/2023/Conference — EMNLP 2023 Findings_

### Official Review · Reviewer_ahUV · 2023-08-04

**Typos Grammar Style And Presentation Improvements:** N/A
**Soundness:** 3

**Excitement:**

3: Ambivalent: It has merits (e.g., it reports state-of-the-art results, the idea is nice), but there are key weaknesses (e.g., it describes incremental work), and it can significantly benefit from another round of revision. However, I won't object to accepting it if my co-reviewers champion it.

**Missing References:**

N/A

**Paper Topic And Main Contributions:**

The paper delves into the incorporation of syntactic knowledge into a language model through additional training. It introduces four pretraining tasks that focus on learning diverse syntactic perspectives. Additionally, the paper explores the utilization of gradient surgery and elastic weight consolidation methods as preventive measures against catastrophic forgetting. The main contributions of this research encompass: (1) the proposition of additional syntactic training incorporating four distinct syntactic tasks for language models, (2) the exploration of gradient surgery and elastic weight consolidation methods to mitigate catastrophic forgetting, and (3) the demonstration of the improved performance of the enhanced model across tasks such as CoLA, RTE, MRPC, and key phrase extraction.



**Questions For The Authors:**

Can you please provide information regarding the average performance in GLUE and whether the new method demonstrates any improvement in this score?

**Reasons To Accept:**

1. The paper extensively investigates the integration of syntactic knowledge into a language model through additional training. It introduces four pretraining tasks that effectively contribute to this goal. The clarity of the ideas presented and the overall ease of following the paper are commendable.
2. The ablation study conducted on four optimizers and four syntactic tasks provides a comprehensive analysis. The inclusion of these experiments sufficiently covers and evaluates the impact of these components.

**Reasons To Reject:**

1. The paper's novelty appears to be somewhat limited, as syntactic training has been widely used in previous methods, and the GS and EWC optimization functions are not entirely new.
2. The study only explores encoder-only language models, and it may have been beneficial to expand the analysis to also include more commonly used decoder-only language models. For instance, it would have been useful to evaluate the proposed approach on models such as GPT.
3. Despite leveraging external syntactic training, the study only demonstrates improvement on four of the tasks in GLUE, with no clear advancements observed in the remaining five tasks.

**Reproducibility:**

3: Could reproduce the results with some difficulty. The settings of parameters are underspecified or subjectively determined; the training/evaluation data are not widely available.

**Reviewer Confidence:**

3: Pretty sure, but there's a chance I missed something. Although I have a good feel for this area in general, I did not carefully check the paper's details, e.g., the math, experimental design, or novelty.

---

> ### Author Rebuttal · Authors · 2023-08-28
>
> We thank you for your comments especially on the GLUE experiments of our paper.
>
> We would like to emphasize the contributions of our paper.
> The main contributions of our paper are twofold: incorporating syntactic knowledge into the model using EWC and GS without changing the structure of the model and analyzing what kind of syntactic knowledge works for what kind of tasks.
> We are confident that these contributions will be also informative when using LLMs other than BERT in the future.
>
>
> For the GLUE task results, the model using GS and EWC was higher than the model using AdamW and SGD for the average score across the eight GLUE tasks. In the comparison between the none model without syntactic pre-training and the model with syntactic pre-training, the effect of syntactic pre-training varied from task to task. This is an important contribution in that it shows that different tasks require different syntactic structures, and it also reveals what tasks require syntactic structures.

---

### Official Review · Reviewer_9i8A · 2023-08-05

**Soundness:** 3

**Excitement:**

4: Strong: This paper deepens the understanding of some phenomenon or lowers the barriers to an existing research direction.

**Missing References:**

Bai et al. (ACL 2021) Syntax-BERT: Improving Pre-trained Transformers with Syntax Trees

**Paper Topic And Main Contributions:**

To improve the peformance of pretrained transformer (BERT) models on tasks that use syntactic knowledge, the authors examine methods to train the model on a syntax-correlated task (dependency relations, phrases, clause status, coordination) together with a specialized optimizer that is intended to mitigate the catastrophic forgetting problem, namely Gradient Surgery or Elastic Weight Consolidation.
The authors find that the syntactic knowledge incorporated in this way improves performance on a subset of the GLUE tasks that they examine as well as for key phrase extraction

**Questions For The Authors:**

(A) What about the other GLUE tasks? Did you just do experiments on the subset presented in the paper?

**Reasons To Accept:**

The method chosen by the authors seems to be fairly general and the results look good

**Reasons To Reject:**

Given that the main contribution consists in reinforcing the syntactic knowledge in a transformer model, the authors could discuss a bit more of the substantial body of literature that tries to do exactly that - on different tasks, or using techniques that are much more cumbersome, etc. - as a reader it isn't clear to me whether relation extraction or some other syntax-heavy task would profit even more from this.

**Reproducibility:**

3: Could reproduce the results with some difficulty. The settings of parameters are underspecified or subjectively determined; the training/evaluation data are not widely available.

**Reviewer Confidence:**

4: Quite sure. I tried to check the important points carefully. It's unlikely, though conceivable, that I missed something that should affect my ratings.

---

> ### Author Rebuttal · Authors · 2023-08-28
>
> Thank you for your comments and pointing out the missing reference.
>
> Many studies (Tian et al., etc.) that embed syntactic structures in language models use relation extraction or similar tasks as the applied task.
> We select GLUE and key phrase extraction as the application tasks because we would like to investigate whether syntactic structures are useful in more general tasks.
> We also include results for the other five GLUE tasks in Appendix, showing that some tasks benefit from the syntactic structure while others do not.
> We think that our paper is worthwhile in that it analyzes in which tasks models with which syntactic structures achieve good performance.

---

### Official Review · Reviewer_TqHP · 2023-08-06

**Soundness:** 3

**Excitement:**

2: Mediocre: This paper makes marginal contributions (vs non-contemporaneous work), so I would rather not see it in the conference.

**Paper Topic And Main Contributions:**

The authors argue that PLMs lack "sufficient" syntactic knowledge, and seek to add this knowledge via fine-tuning. They propose four different syntactic tasks to use during fine-tuning, however, if done naively, fine-tuning on a task which differs from the original pre-training brings the risk of catastrophic forgetting.  Four different methods for optimization (including two which are designed to prevent catastrophic forgetting) are compared on a number of benchmarks from GLUE, and a separate key phrase identification task.  At least one of the catastrophic forgetting-based optimizations typically offer the highest performance on each of the presented GLUE task, and better retain PPL performance.


**Questions For The Authors:**

a)  Is the EWC procedure something first designed in this work?


**Reasons To Accept:**

- Both methods for catastrophic forgetting achieve respectable improvements in performance on GLUE in tasks requiring syntactic knowledge.

- Good experimental rigor
-- Many comparisons
-- Many tasks
-- Each comparison computed across multiple runs/hyperparameter settings

**Reasons To Reject:**

- The PLMs used (BERT) are by current standards, quite old, and quite small.  As work in scaling PLMs up to sizes orders of magnitude greater, performance on syntactic tasks has shown to improve naturally (along with many other useful emergent forms of knowledge).  Some comparison to larger models / application of this method to such models, is necessary to ensure that the method has any practical purpose.

- There are also no baselines from existing work.  There are other forms of fine-tuning, such as adapters, which seek to add additional knowledge to PLMs with less chance of catastrophic forgetting.  The authors even cite one of these papers.  This is also a confusing oversight, because the many appropriate inline citations which contrast various decisions in this work to decisions in existing work demonstrate a great familiarity with the literation, so lacking any comparison to any existing methods is an odd oversight.

- Some questionable design choices.  Perplexity is used as a measure of the model retaining semantic information after fine-tuning, and while that does relate to the original task, there are also aspects of domain drift which are possible and separate from catastrophic forgetting.  How are such factors controlled?

- Related, there is questionable motivation.  Often when talking about catastrophic forgetting, both the original training and the new task are both relevant.  This is clear in the context of robotics, where learning a new behavior should not result in hindering the robot from performing existing behaviors.  Is this true in this case?  PPL is almost always not a valuable end goal, and the entire PLM/LLM paradigm is built around this notion of pre-training in whatever way leads to learning useful linguistic representations, before fine-tuning, aligning, or few-shotting the model towards the task the user actually cares about.  If users never care about both tasks in approximately equal measure, than what good is retaining the original model weights which were not pertinent to the target task?

- There's arguably too much going on here.  The focus of the paper aims to be about catastrophic forgetting, but secondary to that, is also the problem of matching the right syntactic fine-tuning task to the right problem.  This is not entirely known a priori, so all possible pairings are explored, but realistically a good guess can be made (as it would likely be if pursued in a more practical setting).  For instance, it is no surprise that the phrase syntax task helps with key phrase identification.  The disadvantage of the exhaustive approach is that it has both distracted from the main takeaway points while cutting into the space available for supporting the main hypothesis.

- No inclusion of baselines from existing work / SOTA on performance tables

- Key extraction F1 results are better than standard optimizers, but negligibly so.

- No discussion of GC vs EWC.  When a priori would you choose which method?  If the paper included only one such method, traditional optimizers would be the best choice in most situations.

**Reproducibility:**

3: Could reproduce the results with some difficulty. The settings of parameters are underspecified or subjectively determined; the training/evaluation data are not widely available.

**Reviewer Confidence:**

4: Quite sure. I tried to check the important points carefully. It's unlikely, though conceivable, that I missed something that should affect my ratings.

**Typos Grammar Style And Presentation Improvements:**

- Cite GS when discussing it.

- L345: Missing appendix reference

---

> ### Author Rebuttal · Authors · 2023-08-28
>
> We appreciate your highly informative and very detailed comments and questions.
> Discussions with you will be helpful to take our paper in a better direction.
>
>
> First, we would like to share the importance of research that embeds syntactic information in BERT and evaluates its performance in application tasks.
> Small language models such as BERT are still widely used in real applications especially where computational resource is limited. Hence we started with BERT to see if our approach works for models of this size. As you pointed out, there are several studies that have examined whether models such as GPT have linguistic information and can answer those questions, and the newer models have improved their performance. https://aclanthology.org/2022.blackboxnlp-1.24.pdf . It is an interesting research direction to verify the effectiveness of our approach for such larger models, but we believe that our paper is significant in that we have revealed what kind of syntactic information embedded in the models improves performance in which application tasks. To the best of our knowledge, our study is the first to address this research question.
>
>
> Regarding your second reason to reject regarding baselines: our objective is to embed syntactic knowledge into a model without any modification of its model structure because, when we apply any method that changes the model structure such as adapters, it would be difficult to see whether the difference in performance comes from the change in structure or additional syntax training itself. Given a model structure, our objective is to confirm that the model benefits from the additional syntax training.
> Considering the purpose of this discussion, we did show the baselines as "none" column in Table 4 and Table 7, without any syntactic additional pre-training.  Thus other rows clarify the effects of multiple types of syntactic features and optimization techniques.
> We would like to update our paper to emphasize this in the camera ready version.
>
>
> Regarding your 3rd and 4th reasons to reject regarding design choices and motivation: thank you for your insightful comments.
> To measure the ability of our approach to adjust the balance of sensitivity to the syntax and semantics of BERT, the metric should be task-agnostic. That is why we chose PPL as such a metric to observe the degree of holding semantics. We agree that there are other metrics better for our purpose and it is interesting to see how domain drift affects the model performance.
>
>
> We would also like to mention the importance of our simultaneous work on both optimization and syntactic pre-training. As we stated earlier, it was not clear what syntactic structures are important for what tasks since there has been no experiment that properly incorporated each syntactic item into the model. (This includes the task you commented on regarding key phrase extraction.) Therefore, it is worthwhile to combine an appropriate optimization method with pre-training of each syntactic structure.
>
>
> Finally, let us reply to your comment on the choice of GS and EWC. As we described in the paper, one major objective of our study is to analyze the (positive) impact of optimization methods to catastrophic forgetting, not to try any possible combination of an optimization method and a task and see which is the best. We agree that choosing different downstream tasks and/or optimization methods may give a slightly different result, but we believe that our findings will shed some light on future progress in incorporation of syntax into LLMs.
>
>
> We would be happy to reflect these discussions to the camera ready version.

---

### Meta-Review · Area_Chair_sSWW · 2023-09-17

**Recommendation:** 4

**Metareview:**

The paper provides solid experiments over a variety of settings, making a convincing case quantitatively for the claim of improvement over the baseline that does not include training with explicit syntactic knowledge. An issue brought up in each of the three reviews is proper placement of the proposed approach in the context of existing work on adding syntactic knowledge of language models. Although direct quantitative comparisons may not be straightforward, the authors can still present a qualitative case for the approach, contrasting it more directly with existing work in greater detail, since the reviewers identified this as a weakness of the paper. A second issue is whether the approach would be effective with other kinds of models, especially those in more current use, but this is beyond the scope of the current paper. Overall, the paper presents interesting results that add to the understanding of a kind of language model that may not be the most current, but was widely used in recent years and is still relevant.

---

### Decision · Program_Chairs · 2023-10-07

**Decision:**

Accept-Findings

**Comment:**

The paper provides solid experiments over a variety of settings, making a convincing case quantitatively for the claim of improvement over the baseline that does not include training with explicit syntactic knowledge. An issue brought up in each of the three reviews is proper placement of the proposed approach in the context of existing work on adding syntactic knowledge of language models. Although direct quantitative comparisons may not be straightforward, the authors can still present a qualitative case for the approach, contrasting it more directly with existing work in greater detail, since the reviewers identified this as a weakness of the paper. A second issue is whether the approach would be effective with other kinds of models, especially those in more current use, but this is beyond the scope of the current paper. Overall, the paper presents interesting results that add to the understanding of a kind of language model that may not be the most current, but was widely used in recent years and is still relevant.